# Rampant Interkingdom Horizontal Gene Transfer in Pezizomycotina? An Updated Inspection of Anomalous Phylogenies

**DOI:** 10.3390/ijms26051795

**Published:** 2025-02-20

**Authors:** Kevin Aguirre-Carvajal, Sebastián Cárdenas, Cristian R. Munteanu, Vinicio Armijos-Jaramillo

**Affiliations:** 1Computer Science Faculty, University of A Coruna, CITIC-Research Center of Information and Communication Technologies, 15071 A Coruña, Spain; kevin.aguirre@udla.edu.ec (K.A.-C.); c.munteanu@udc.es (C.R.M.); 2Bio-Cheminformatics Research Group, Universidad de Las Américas, Quito 170513, Ecuador; 3Carrera de Ingeniería en Biotecnología, Facultad de Ingeniería y Ciencias Aplicadas, Universidad de Las Américas, Quito 170513, Ecuador; julian.cardenas@udla.edu.ec

**Keywords:** interkingdom horizontal gene transfer, anomalous phylogenies, Pezizomycotina

## Abstract

Horizontal gene transfer (HGT) is a significant source of diversity in prokaryotes and a key factor in their genome evolution. Although similar processes have been postulated for eukaryotes, the validity of HGT’s impact remains contested, particularly between long-distance-related organisms like those from different kingdoms. Among eukaryotes, the fungal subphylum Pezizomycotina has been frequently cited in the literature for experiencing HGT events, with over 600 publications on the subject. The proteomes of 421 Pezizomycotina species were meticulously examined to identify potential instances of interkingdom HGT. Furthermore, the phylogenies of over 275 HGT candidates previously reported were revisited. Manual scrutiny of 521 anomalous phylogenies revealed that only 1.5% display patterns indicative of interkingdom HGT. Moreover, novel interkingdom HGT searches within Pezizomycotina yielded few new contenders, casting doubt on the prevalence of such events within this subphylum. Although the detailed examination of phylogenies suggested interkingdom HGT, the evidence for lateral gene transfer is not conclusive. The findings suggest that expanding the number of homologous sequences could uncover vertical inheritance patterns that have been misclassified as HGT. Consequently, this research supports the notion that interkingdom HGT may be an extraordinary occurrence rather than a significant evolutionary driver in eukaryotic genomes.

## 1. Introduction

Horizontal gene transfer (also called lateral gene transfer) is a well-established phenomenon in prokaryotes, facilitating the assimilation of exogenous genetic material and increasing population variability [1,2]. However, the role of HGT in eukaryotes is not as well established. With new genomic data available, several reports of HGT in eukaryotes, including interkingdom HGT, have emerged [3,4,5,6,7,8,9,10]. These events seem to provide recipient species with new adaptations for their environments. This has been notably observed in the subphylum Pezizomycotina, a group of filamentous fungi comprising most ascomycetes. For instance, Zhang et al. [11] reported that the emergence of the fungal entomopathogen *Metharizium* was facilitated by horizontally transferred genes. Yin et al. [12] provided evidence of HGT events from bacteria and eukaryotes driving the adaptation of the apple pathogen *Valsa mali*. Furthermore, Marcet-Houben and Gabaldón [13] observed a widespread presence of bacterial and plant genes in Pezizomycotina and other fungal genomes, hypothesizing significant roles for these genes in the host species. In total, the literature review uncovered over seventy papers documenting instances of interkingdom HGT in Pezizomycotina species (see Appendix A).

Despite abundant evidence, the apparent ease with which genes are transferred and retained in Pezizomycotina genomes is puzzling. Eukaryotes possess numerous mechanisms that guard against lateral gene transfer, such as nuclear membranes, specific intron processing, and gene-silencing mechanisms for unpaired DNA [14,15,16,17]. Could interkingdom HGT be a relevant source of variability for eukaryotes? Might there be a hidden mechanism allowing Pezizomycotina to capture and integrate foreign DNA?

To address these questions, it is essential to distinguish interkingdom HGT from other evolutionary processes. Common confounders include incomplete lineage sorting, differential gene loss, variable mutation rates, and erroneous phylogenies, which could be mistaken for HGT signals [18]. Several discussions have questioned the reliability of gene-by-gene phylogenies for detecting HGT, particularly when comparing unrelated species [19,20]. Others, however, consider this an accurate method [21,22]. Beyond detection methods, some researchers argue against the widespread acceptance of interkingdom HGT in eukaryotes, suggesting that lateral gene transfer should be attributed primarily to the genes transferred by the bacteria that led to mitochondria and plastids [19,20,23]. They point to a lack of evidence for cumulative evolutionary effects, akin to those seen in prokaryotes, and a lack of consistent pangenomes. Conversely, multiple studies support the occurrence of interkingdom HGT in multicellular eukaryotes, proposing intricate mechanisms for this process [24,25,26].

To assess the impact of interkingdom HGT in Pezizomycotina, we performed a comprehensive de novo analysis to identify potential HGT candidates across all available Pezizomycotina proteomes. Additionally, we gathered and reexamined published data on reported interkingdom HGT candidates, integrating the latest genomic information. All the collected data were carefully analyzed using phylogenetic methods and well-defined criteria to distinguish HGT patterns from other anomalous phylogenies. Our study aimed to provide an up-to-date, data-driven perspective on the prevalence of interkingdom HGT in Pezizomycotina species.

## 2. Results

In this study, a meticulous approach was employed to identify interkingdom horizontal gene transfer in the subphylum Pezizomycotina through phylogenetic reconstruction. To ensure comprehensive sampling of the subphylum, complete proteomes were obtained from the UniProt database. Initially, an automated tool (DarkHorse 2) was utilized to identify phylogenetically unusual proteins (with a high percentage of homologues in kingdoms outside the fungi) within the proteomes. This process identified 246 candidates in 421 proteomes based on an LPI (lineage probability index) of less than 0.7 and best hits that were different from fungi. The identified proteins were carefully analyzed using multiple steps of phylogenetic reconstruction to minimize errors in tree inferences.

Multiple groups of potential homologues (xenologous) for each candidate were examined to assess the consistency of HGT patterns observed in trees. Consistent repetitive patterns were observed throughout the analysis (Figure 1, Appendix A). Subsequently, only one candidate met the criteria for an HGT case (designated as HGT 1—Appendix A; Appendix A). This involved either a single sequence or a group of Pezizomycotina proteins positioned within a non-fungal clade without evidence of contamination (Figure 1A).

For HGT 1, it was also noted that a fungal protein was surrounded by bacterial homologues in the phylogenetic tree reconstruction. Additionally, examination revealed that the gene encoding this protein was inserted into a contig with upstream and downstream sections showing no signs of HGT or indicating suspicious contamination during sequencing processes. Another noteworthy characteristic specific to HGT1 is its possession of InterPro domain HD phosphohydrolase, reflected as Phosphopantetheine adenylyltransferase (PTHR21174:SF0) in the Panther database, making it unique among all sequences found in the *Massariosphaeria phaeospora* genome.

In contrast, 134 candidates were discarded as potential cases of HGT because their topology indicated a vertical transmission scenario, meaning the fungal sequences formed a single, cohesive monophyletic group, while the non-fungal sequences occupied a distinct, separate clade (see Figure 1B). This group includes cases that resemble HGT and are associated with a highly conserved domain, such as a ubiquitin-like domain, which is found in multiple kingdoms with high conservation levels. During the search using DarkHorse, no fungal hits were recovered for these potential HGT candidates at the protein level through BLASTp; however, when the same search was conducted at the DNA level using BLASTn, fungal BLAST hits were obtained for these candidates. This phenomenon was observed in proteins AEO55424, EUC31191, EMD85945, and EMD93446.

Ten candidates displayed a similar pattern of HGT but including fungal homologues in different phyla (Figure 1C). In order to elucidate the evolution of these proteins, a hypothesis of a very ancient transfer at the origin of fungi must be proposed, followed by gene loss in most fungal species. Under this framework, alternative explanations may account for this pattern better than HGT does. Furthermore, 101 candidates were identified without homologues in the fungal kingdom. To identify potential contamination, the scaffolds from which they originated in their respective sequencing projects were analyzed (see Section 4). The findings reveal that these candidates are situated on contigs with a bacterial origin. These HGT patterns are attributed to contamination in the sequencing projects (Figure 1D).

In the first analysis of interkingdom HGT cases, a small number of potential candidates were found. Intrigued by this outcome, the approach was replicated using documented HGT cases in the scientific literature. Examination of interkingdom HGT reports for Pezizomycotina yielded 39 articles reporting instances of HGT within this species group (see Appendix A). These articles provided us with 275 candidate genes (see Appendix A). The results from these candidates displayed similar patterns as shown in Figure 1. Among them, seven exhibited HGT patterns (Figure 1A, Appendix A; Appendix A). It was observed that 243 trees showed a topology consistent with vertical transference rather than lateral transference (Figure 1B). For 21 candidates, an ancient transfer event is required—possibly originating at the root of Pezizomycotina or even further back—followed by gene loss across most current species to justify an HGT scenario (Figure 1C). Additionally, two candidates indicated signs of contamination on their contigs (Figure 1D). Lastly, for the remaining two candidates, only a few similar sequences were found in the databases, making it impossible to reconstruct an informative phylogenetic tree due to a lack of homologues. The phylogenetic trees and multiple sequence alignments are available for download at the Zenodo repository (https://zenodo.org/records/14455232 (17 February 2025)).

After identifying candidates from the conducted searches (following the pattern in Figure 1A), the putative horizontally transferred genes within the species tree were determined to locate the minimum point of transfer for each case. Out of eight cases, three potential HGT candidates were detected in the Aspergillaceae family (HGT 4–6) and one in the Valsaceae family. The ancient transfer was identified in the Chaetothyriales order, followed by candidates found in the *Fusarium* genus (HGT 7) and exclusively within *Massariosphaeria phaeospora* species (HGT 1). All these cases have a putative origin in bacteria. A single candidate with potential HGT from plants was also found, linked to the *Botrytis* genus (HGT 8).

At the same time, the role of the candidate proteins and their association with the lifestyle of the presumed receptor species were examined. However, in some instances, there is insufficient information about these proteins, and it is unclear how relevant they are to potential HGT candidates for the receptor species. One example is OQE16981 from *Penicillium steckii*, which has an unidentified function (annotated as a hypothetical protein without detectable InterPro domains), or the histidine–aspartate (HD)-domain protein (HGT 1). Nonetheless, at least two candidates were identified that might have significant functions for accommodating receptor lifestyles, such as the SUKH family, a group of proteins involved in bacterial toxin systems serving as immunity factors, or thiosulfate sulfurtransferase, which is involved in cyanide detoxification and could be an important protein for plant pathogens. The complete list of protein annotations and fungal lifestyles can be found in Table 1 and Appendix A.

## 3. Discussion

In this work, interkingdom HGT instances were investigated in complete proteomes of Pezizomycotina. A traditional approach of reconstructing protein-by-protein trees was utilized to identify phylogenetic anomalies before applying a BLAST-based proteomic HGT prediction tool. Upon following this conventional method, only one potential candidate exhibiting the characteristics of an HGT candidate in a phylogenetic tree was identified. Intrigued by this outcome, the reported HGT cases were examined, and a similar pattern was observed; only 2.5% of the analyzed candidates met the criteria for an HGT pattern. At this juncture, two possible explanations emerged for the results: (1) the methodology may be influencing the observations, or (2) the expansion of databases (particularly in eukaryotic genomes) is revealing fungal homologues that were previously unavailable.

Acknowledging the limitations of the methodology, including redundancy in prokaryotic homologues and the inability to distinguish between HGT and ancient paralogy plus differential gene loss [3,20,27], BLAST-based methods have faced criticism for overestimating the number of HGT candidates [19]. In contrast, a low proportion of plausible cases was observed from our approach. This suggests that the reduction in HGT candidates is not solely due to the method employed but also to the increased availability of new fungal homologues. A similar pattern was seen in early reports on the human genome [28], which initially showed a high proportion of foreign genes but diminished as more genomes were involved in analysis [29,30]. A recent study examined the impact of database composition on the detection of interdomain HGTs in Pezizomycotina. The findings indicate that the increased representation of eukaryotic sequences in databases has significantly decreased the number of interdomain HGT candidates identified within the fungal phylum Pezizomycotina [31].

The phylogenetic patterns depicted in Figure 1 reflect the most frequent findings from the analysis of 521 manually assessed trees conducted in this study. In the de novo detection approach, a significant number of excluded cases stemmed from potential contamination within genome assemblies. This outcome was anticipated due to the rapid expansion of genome sequencing activities and database sizes [32]. Another recurrent pattern involved the identification of horizontally transferred gene candidates within a clade comprising sequences from distinct classes of Ascomycota or, in some instances, different fungal phyla, such as Basidiomycota or Chytridiomycota. The recognition of these patterns as HGT events suggests long-term transfer at the root level of Ascomycota or fungi followed by loss in most descendant lineages. A more plausible explanation would be that these genes were present in ancestral species with subsequent gene loss and no lateral transfer involved, as discussed extensively by Bremer et al. [27]. Supporting this notion, various identified candidates exhibiting this pattern yielded similar sequences through a simple BLAST search across different eukaryotic kingdoms. It is also conceivable that, with future advancements leading to increased information availability or improved gene annotation inference, homologous sequences for these candidates may reappear in lineages currently perceived as lost [33,34].

Most of the studied phylogenies (predicted as HGT) generally follow a vertical transmission topology (Figure 1B), which suggests that the tool used for de novo detection may lack specificity, particularly for Pezizomycotina species. When it comes to bibliographic cases, it is argued that new information and increased availability of homologous sequences for fungi can restructure anomalous phylogenies that initially appeared as interkingdom HGT cases in smaller databases. Salzberg et al. [29] tested this effect on human genome HGT candidates and Aguirre-Carvajal et al. [31] in Pezizomycotina.

Eight instances of interkingdom HGT were identified. These occurrences indicate lineage-specific transfers that may have originated from either bacteria (HGT 1–7) or plants (HGT 8). No evidence of contamination was observed for these cases; however, the lack of homologous sequences in eukaryotes is remarkable. If true, these candidates represent punctual insertions in the Pezizomycotina species tree (Figure 2 and Appendix A), yet most of the candidates show redundant functions in donor genomes (other proteins with similar annotation exist within the same species), suggesting a lack of novelty in the putative transferred genes. One exception to this is the HGT 4, which contains a SUKH-3 immunity protein, a bacterial molecule implied in the toxin defense mechanisms [35]. Curiously, according to the InterPro database (https://www.ebi.ac.uk/interpro/, accessed 14 November 2023), the SUKH-3 domain (IPR025850) has only been observed in bacteria and *Penicillium* and *Aspergillus* species. The relevance and evolution of these proteins could be interesting to investigate in the future. Another exception is that HGT 1 contains the HD phosphohydrolase domain (IPR009218). This domain was not found in any other *Massariosphaeria phaeospora* protein, but it is present in other fungi, Metazoa, archaea, and viruses, suggesting that fungal homologues could be identified in the future.

Alternative explanations for HGT in phylogenetic anomalies include extreme mutation rates, lineage gene loss, or segregating paralogous genes [16,19]. Evaluating interkingdom HGT reveals that lineage gene loss to an entire kingdom maintained only in specific members of a genus or family is an implausible explanation. However, the research by Bremer et al. [27] suggests that lineage gene loss may occur more frequently than previously anticipated. Extreme mutation rates cannot account for the similarity between fungal and non-fungal sequences. Few similarities between fungal proteins and non-fungal ones were expected; however, it was observed that our candidates were more similar to non-fungal sequences. Regarding the results of this study, the relevance of interkingdom HGT patterns could be reduced to the presence/absence/identification of homologous (both orthologous and paralogous) sequences available to reconstruct phylogenies. The presence of more sequence availability decreases the chances of finding HGT-compatible phylogenies. Therefore, it is advisable, when proclaiming the presence of an HGT candidate, to ensure there are no homologous species showing a vertical transmission scenario. Based on our claim’s rationale, future analyses using the same data may find vertically congruent patterns rather than the HGT candidates reported here.

In eukaryotic organisms, there are numerous obstacles to the transmission of genetic material, particularly in multicellular lineages. These barriers include gene-silencing mechanisms for unpaired DNA and different codon-utilization biases [14,36]. Then, assuming that the genetic material overpasses these barriers, it must then spread through the population and become fixed in the entire species, or at least, the ancestral gene pool should come from the part of the population that contains the HGT event. Given this context, it is difficult to conceive a rampant flux of genes across kingdoms. Despite numerous publications on this topic in recent years, the mechanisms for transferring and fixing foreign genetic material within populations have not been firmly established [37]. In fact, the fixation of interkingdom HGT in eukaryotes goes against probabilities. This line of reasoning leads to the conclusion that interkingdom HGT in eukaryotes is probably an event that is more isolated than is recurrent. The results obtained from this study support this perspective at least for Pezizomycotina.

The utilization of comprehensive approaches (e.g., Cote-L’Heureux et al. [3]) for detecting interkingdom HGT may be insufficient if there are missing homologous genes that reveal vertical transmission. In eukaryotes, particularly in fungi, genome completeness and availability remain limited [33,38,39,40]. This should prompt careful consideration of candidates for interkingdom HGT. Future reports about candidates should carefully consider alternative hypotheses and adopt highly conservative pipelines for interkingdom HGT detection to avoid conducting in-depth analysis of spurious genes or genome fractions.

## 4. Materials and Methods

### 4.1. Interkingdom HGT Protein Candidates’ Detection

To identify potential proteins that may have undergone interkingdom HGT into Pezizomycotina, two methods were employed. First, Darkhorse 2.0 software [41] was utilized to mine proteome data and infer potential interkingdom HGT events (de novo approach). Second, a comprehensive literature search was conducted in the PUBMED engine to extract proteins that have been previously reported as HGT candidates in Pezizomycotina (bibliographic search). In both cases, individual phylogenies were reconstructed for each candidate, and each tree was manually evaluated in order to detect topologies compatible with interkingdom HGT events.

Darkhorse 2.0 is a computational tool utilized to predict potential instances of horizontal gene transfer within proteomes. The approach employs a probabilistic framework that considers the relationships between query proteins and their database matches in terms of phylogenetic relatedness. The software uses BLAST 2.15.0 alignments to first identify putative orthologous prior to performing lineage probability index (LPI) calculations—a metric designed to detect candidates for horizontal gene transfer. The program retrieves lineage data from the taxonomy database for potential matches and computes preliminary estimates of the likelihood of phylogenetic inheritance, which serve as probabilistic guidelines for the ranking process. Subsequently, a second iteration of LPI calculations refines the ranking to identify the candidates with higher probabilities of being evolutionarily related between query sequence and their BLAST hits. When calculating LPI, the DarkHorse algorithm examines the taxonomic distribution of the top BLAST hits and compares the taxonomic distance of these hits to the query sequence’s taxonomic annotation. A low LPI score suggests that the majority of hits come from evolutionarily distant species, supporting the hypothesis of horizontal gene transfer. Conversely, a high LPI indicates that the sequence is more likely to have been inherited through vertical descent. The fast and automated capabilities of Darkhorse 2 enable efficient analysis of entire proteomes, rendering it well suited for comprehensive screening of large datasets within a single application.

The NCBI non-redundant (NR) protein database and associated taxonomic data were used, including the names.dmp, nodes.dmp, and prot.accession2taxid files, to construct a reference protein dataset using the Diamond v2.1.9 tool [42]. Subsequently, a MySQL database was created containing this compiled NR protein dataset.

The complete Pezizomycotina proteomes, obtained from the UniProt database [43] in February 2022 (Appendix A), were compiled. A Diamond BLASTp v2.1.9 search was then conducted against the NR database using these proteomes. Subsequently, the BLASTp hit results were used as input for the Darkhorse 2.0 software, applying the following parameters: a maximum line per packet of 4000, a minimum of 3 lineage terms, and a minimum alignment coverage of 0.7. To focus on interkingdom horizontal gene transfer (HGT), candidates with local LPI values below 0.7 and BLAST hits originating from species outside the fungal kingdom were specifically targeted. This rigorous selection process led to the identification of 247 potential interkingdom HGT candidates for further analysis.

To complement the computational approach, a comprehensive review of the literature was conducted, with the specific objective of identifying instances of interkingdom HGT within the fungal subphylum Pezizomycotina. This search process involved the use of targeted keywords, such as “Interkingdom”, “Fungi”, “Pezizomycotina”, “Horizontal Gene Transfer”, and “Lateral Gene Transfer”, in the PubMed advanced search engine. The details of these searches, along with the number of papers obtained, are presented in Table 2. After removing any duplicate entries, a total of 618 relevant articles were compiled.

Following a rigorous analysis, articles were included if they documented at least one case of interkingdom HGT in a Pezizomycotina species, the candidate protein was identifiable in a public database or provided by the authors, and the article was written in English. Conversely, articles without abstracts referencing HGT cases, not involving interkingdom candidates, those where HGT occurred outside the Pezizomycotina subphylum, or those with duplicate candidates were excluded (Appendix A displays the detailed inclusion and exclusion analysis with the selected papers). Ultimately, 39 articles were selected that collectively identified 275 candidate proteins participating in interkingdom HGT events in Pezizomycotina.

### 4.2. Phylogenetic Analysis for Interkingdom HGT Detection

From the NCBI database, a dataset comprising 521 proteins in GenPept format was assembled, which had been previously recognized through Darkhorse 2.0 (246 proteins) and a literature review (275 proteins). Each protein underwent a comprehensive extensive processing pipeline (Figure 3). Initially, the amino acid sequences in FASTA format were extracted from the corresponding GenBank files using the BIOPYTHON v1.78 library [44]. Subsequently, each sequence was employed as a query in a local Diamond BLASTp v2.1.9 search against the NR dataset previously compiled. Two distinct BLASTp runs were performed using “taxonlist” command: one restricted to the fungi taxon (fungi txid:4751) and another specific to the putative source of the transferred gene (donor species), encompassing bacteria (bacteria txid:2), metazoa (metazoa txid:33208), and plants (viridiplantae txid:33090). The putative donor species was identified in the Darkhorse analysis or from literature reference.

To identify similar sequences within Pezizomycotina genomes, specific filter criteria were applied to refine the BLAST results. This included a pairwise similarity threshold of over 35%, query coverage above 65%, and an e-value below 10^−5^. The NCBI EFETCH v16.2 tool was then used to retrieve the sequences that met these parameters.

When candidate genes lacked fungal counterparts, an additional curation process was performed to detect potential contamination within the contigs harboring the candidate sequences. For candidate genes located in small contigs, BLAST searches were conducted on the surrounding intergenic and coding regions upstream and downstream of the gene. If the entire contig demonstrated superior BLAST matches to sequences from a non-fungal kingdom, the candidate was excluded due to presumed contamination.

In order to mitigate redundancy within the dataset, the CD-HIT v4.8.1 program [45] was employed with a 100% similarity threshold to remove identical sequences while retaining important taxonomic data. This was essential to minimize potential biases and inaccuracies in the subsequent phylogenetic analyses. Duplicate sequences have the capacity to erroneously influence clade support, potentially leading to flawed biological conclusions.

The non-duplicated sequences obtained were combined with the query sequence to maintain the complete set of homologues. The MAFFT v7.520 tool [46] was used for aligning amino acid sequences, which was then refined using GBLOCKS v0.91b software [47], employing their default settings.

The modeltest-ng v0.1.7 software [48] was employed to select the most appropriate model for each sequence alignment. A maximum likelihood phylogenetic tree was constructed using PHYML v 3.3 [49], incorporating Shimodaira–Hasegawa support values and generating the tree output in Newick format.

After reconstructing the phylogenetic trees of each candidate, the taxonomic information for the set of homologues was extracted from NCBI using custom Python 3.10.12 scripts. These data were then used to generate phylogenetic trees in nexus format, which were manually analyzed using Geneious Prime 2023.2.1 (https://www.geneious.com (17 February 2025)) to identify instances of HGT.

A critical analysis of the tree topologies was conducted to assess the plausibility of the HGT hypothesis. Certain patterns were ruled out as cases of interkingdom HGT. First, when the HGT candidate is clustered with multiple Pezizomycotina species in a monophyletic group, while the non-fungal homologues form a separate cluster, this scenario is difficult to reconcile with HGT and is consistent with vertical transmission (Figure 1B). The second case dismissed involved the HGT candidate and a few Pezizomycotina sequences being clustered with sequences from diverse fungal phyla, such as Basidiomycota, Mucoromycota, or Chytridiomycota, as well as sequences of non-fungal origin (Figure 1C). To consider this pattern as interkingdom HGT, it would be necessary to accept the transfer occurring near the origin of the fungal kingdom, followed by differential gene loss in the majority of fungal descendants. Alternatively, it could be explained by the differential loss of the genes in most current fungal species, with the gene remaining similarity in homologues outside the fungal kingdom. The HGT hypothesis would require an additional step of interkingdom transfer to account for these tree topologies; consequently, the premature loss of the gene within the fungal kingdom represents a more parsimonious explanation.

A third scenario that was ruled out as an instance of interkingdom HGT was when the candidate lacked homologues within the fungal kingdom (Figure 1D). All such sequences were thoroughly examined for potential contamination (following the procedure mentioned above). Candidates that did not pass the contamination assessment were subsequently excluded from consideration as HGT cases.

Plausible candidates for interkingdom HGT were defined as those instances in which reconstructed phylogenetic trees showed one or more Pezizomycotina sequences clustering with non-fungal proteins without evidence of additional fungal homologues in the databases (Figure 1A).

An overview of the research methodology can be found in Figure 3.

### 4.3. Species Tree Reconstruction

For the reconstruction of the species tree for Pezizomycotina species, 586 genomes and six phylogenetic markers (P00927, P05694, P38132, P38791, P47079, and Q02892) from the Universal Fungal Core Genes (https://ufcg.steineggerlab.com (17 February 2025)) were utilized. Each marker was retrieved from all species through a BLAST search that selected results with an e-value below 0.001, over 40% identity percentage, and more than 70% query coverage for further analysis. The concatenated sequences of each species’ markers underwent MAFFT alignment, followed by the construction of a species tree using PHYML v 3.3. Taxonomic annotation on the labels of the phylogenetic tree was carried out in RStudio 2022.12.0 using the ‘ape’ package v5.7 [50] to improve result interpretation. A rigorous evaluation of this tree was performed using Geneious Prime 2023.2.1.

## 5. Conclusions

The extensive documentation of interkingdom HGT in Pezizomycotina species suggests that the lateral transfer of genes is a prevalent source of variability in these organisms. However, upon careful examination, it becomes clear that this phenomenon may be more limited than previously thought. Out of 521 analyzed trees with expectations for lateral gene transfer, only eight phylogenies were found to be compatible with interkingdom HGT. This unexpected outcome raises questions about the significance of HGT as a driving force in Pezizomycotina evolution and calls into doubt the reliability of previous reports on HGT. The analysis indicates that the absence of homologous sequences increases the likelihood of observing anomalous phylogenies that might mistakenly be interpreted as evidence for HGT events. Furthermore, updated versions show patterns declared as cases of past HGT now aligning with vertical compatibility. Thus, caution should be exercised when interpreting reports based on phylogenetic data due to the challenges associated with reconstruction and evolutionary processes that produce anomalous phylogenies pending further expansion from databases.

## Figures and Tables

**Figure 1 ijms-26-01795-f001:**
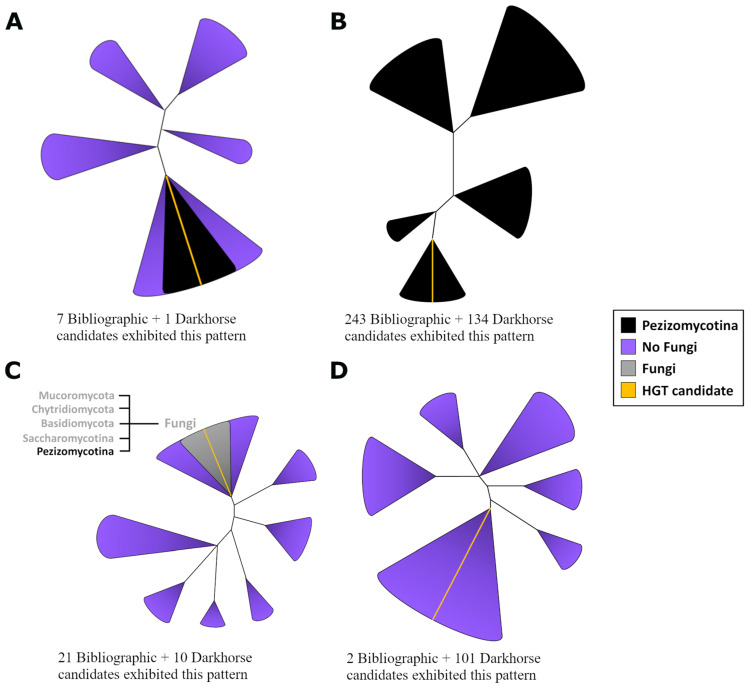
Diagram illustrating various unrooted phylogenetic tree patterns observed during the search for horizontal gene transfer candidates. (**A**) The pattern expected in interkingdom HGT events. Here, the candidate and a group of homologues from the Pezizomycotina subphylum (in the same clade) were observed without more detectable homologues in eukaryotes. (**B**) The patterns expected in vertical transference with the candidate clustered with homologues from Pezizomycotina. In this pattern, it is common to observe some bacterial or plant sequences included from the BLAST searches. However, the non-fungal sequences cluster in a monophyletic group, which is consistent with an expected vertical transfer pattern. (**C**) This pattern reflects the presence of few homologues in fungi, surrounded by no fungal homologues; these fungal homologues came from distantly related classes or phyla. To accept this pattern as an HGT case, the transference should happen at or near to the fungal ancestor followed by the loss of the gene in the most part of the species of the kingdom. (**D**) This pattern reflects only one fungal sequence surrounded by no fungal homologous sequences; the candidates that follow this pattern were scrutinized as potential contamination.

**Figure 2 ijms-26-01795-f002:**
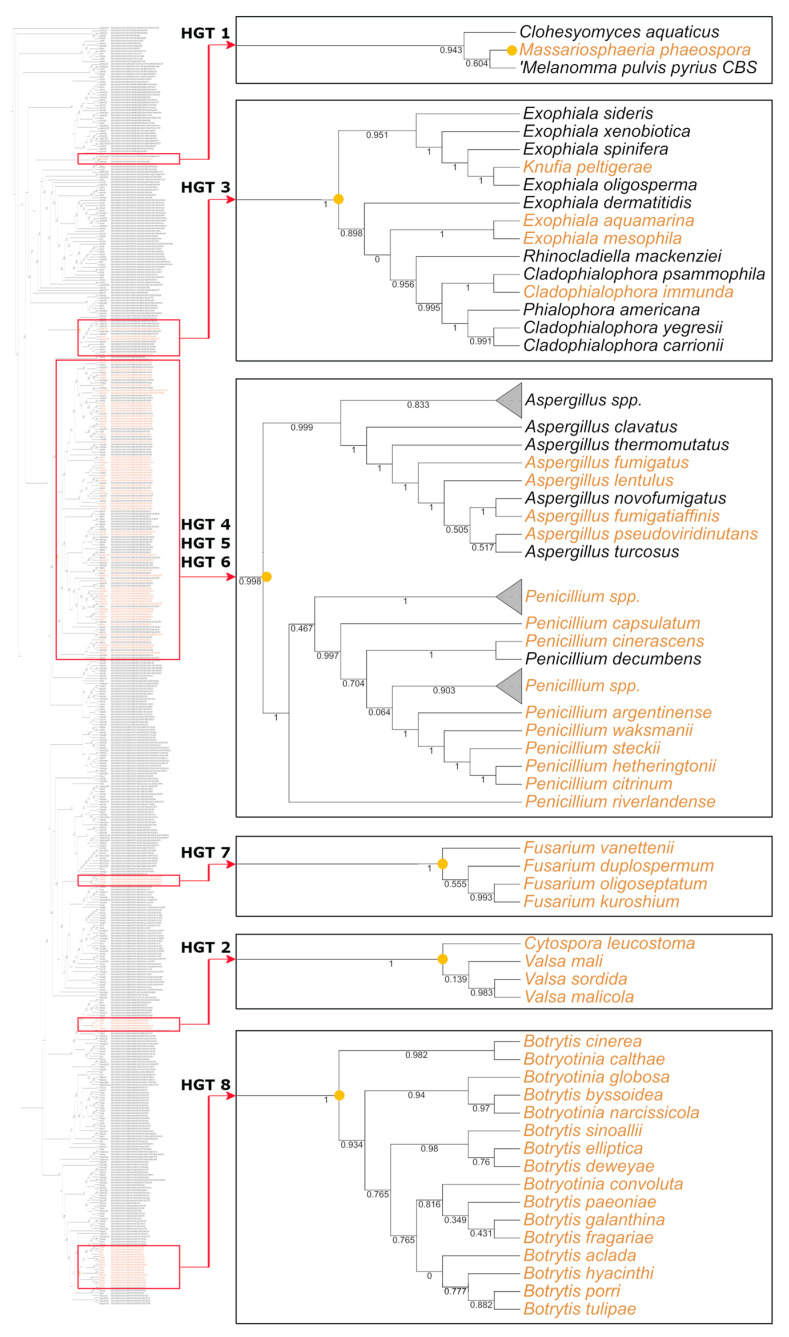
Species tree depicting horizontal gene transfer (HGT) events within Pezizomycotina. Black squares connect to close-up views of specific locations relevant to HGT events in the species tree. Yellow labels indicate species involved in HGT events (the specific protein tree for each HGT case can be observed in the Appendix A). These species represent putative receptor organisms of the HGT events. Yellow circles (internal nodes) indicate the minimum common ancestor inferred for the transference. Eight HGT instances were identified in Pezizomycota species: *Massariosphaeria phaeospora* (HGT 1), Valsaceae species (HGT 2), Chaetothyriales species (HGT 3), Aspergillaceae (HGT 4, 5, and 6), *Fusarium* genus (HGT 7), and *Botritys* genus (HGT 8). The complete tree is shown in Appendix A. The values along the branches represent Shimodaira–Hasegawa support values.

**Figure 3 ijms-26-01795-f003:**
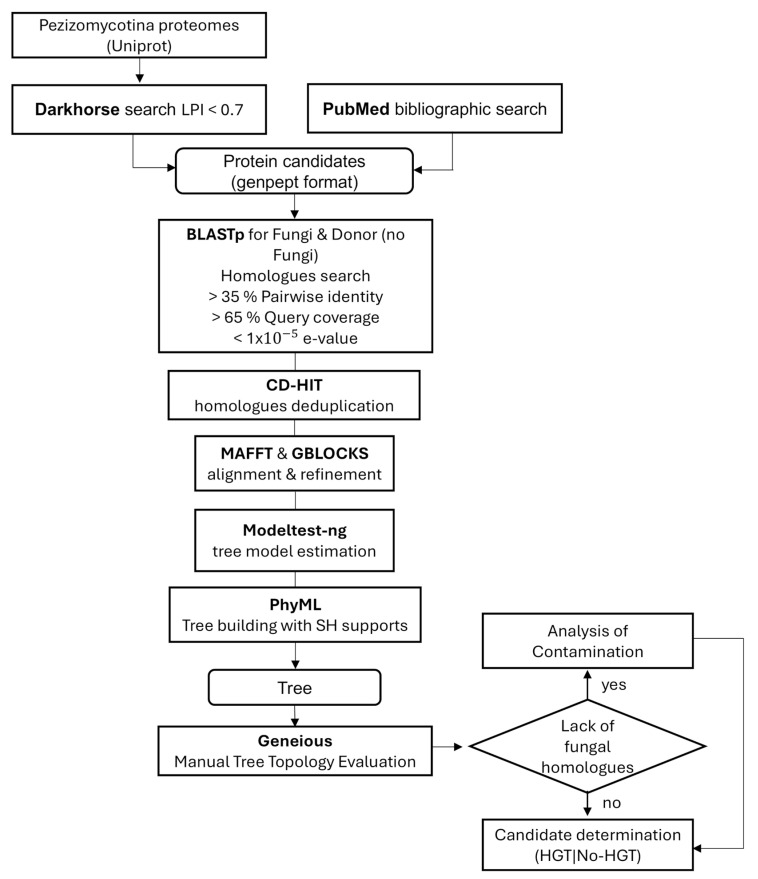
Workflow employed in this investigation for the detection of horizontal gene transfer candidates.

**Table 1 ijms-26-01795-t001:** List of observed HGT congruent candidates in this study.

HGT Event *	Protein Annotation	Putative Function	Receptor Organisms	Lifestyle
HGT 1	HD domain-containing protein	-	*Massariosphaeria phaeospora*	Saprophytic
HGT 2	Thiosulfate sulfurtransferase	Proteins involved in cyanide detoxification. Cyanide compounds are used in plant defense	Valsaceae family species	Phytopatogen
HGT 3	Alcohol dehydrogenase GroES-like domain-containing protein	Several roles. Potential toluene degradation	Chaetothyriales order species	Saprophytic/Opportunistic animal pathogens
HGT 4	SUKH-3 immunity protein	Potential immunity proteins for a diversity of toxin systems	Aspergilleacea family species	Mainly Saprophytic
HGT 5	Hypothetical protein	-	Aspergilleacea family species	Mainly Saprophytic
HGT 6	LabA like, NYN domain containing protein	RNA binding, potentially with a ribonuclease activity	Aspergilleacea family species	Mainly Saprophytic
HGT 7	Rossmann-fold NAD(P)(+)-binding proteins	Involved in several processes (e.g., redox reactions and electron transfer)	*Fusarium* genus	Saprophytic/plant pathogen
HGT 8	Glycosyltransferase family 1	Catalyze the transfer of glycosyl residues from their specific donor to an acceptor molecule	*Botrytis genus*	Mainly plant pathogens

* Appendix A show the complete list of proteins involved in each HGT event.

**Table 2 ijms-26-01795-t002:** Searches conducted in Pubmed to retrieve publications on horizontal gene transfer in Pezizomycotina (8 August 2023).

Search Line	Number of Results
“horizontal gene transfer”[Title/Abstract] OR “lateral gene transfer”[Title/Abstract] AND interkingdom[Title/Abstract] AND Fungi[Title/Abstract]	10
“horizontal gene transfer”[Title/Abstract] OR “lateral gene transfer”[Title/Abstract] AND Fungi[Title/Abstract]	471
Fungi[Title/Abstract] AND Bacteria[Title/Abstract] AND HGT[Title/Abstract] AND “Horizontal gene transfer”[Title/Abstract] AND Interdomain	4
Pezizomycotina[Title/Abstract] AND Bacteria[Title/Abstract] AND HGT[Title/Abstract] AND “Horizontal gene transfer”[Title/Abstract] AND Interdomain	1
(“horizontal gene transfer”[Title/Abstract] OR “lateral gene transfer”[Title/Abstract]) AND (“Fung*”[Title/Abstract])	617
(“horizontal gene transfer”[Title/Abstract] OR “lateral gene transfer”[Title/Abstract]) AND (“fung*”[Title/Abstract] OR “Pezizomycotina”[Title/Abstract])	618
(“horizontal gene transfer”[Title/Abstract] OR “lateral gene transfer”[Title/Abstract]) AND (Pezizomycotina[Title/Abstract])	21

## Data Availability

Additional data are available in Appendix A and in the Zenodo repository https://zenodo.org/records/14455232 (17 February 2025).

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
