# Peer review of "Rampant Interkingdom Horizontal Gene Transfer in Pezizomycotina? An Updated Inspection of Anomalous Phylogenies"

_ijms, 2025, doi:10.3390/ijms26051795_

Round 1
Reviewer 1 Report
Comments and Suggestions for Authors
(1) in the abstract, the authors should describe the study and results with past tense, instead of present tense.
(2) I wonder why the authors used proteomes data to search HGT, rather than the genome data.
(3) The study did not clarify the origin on the HGTs in Pezizomycotina fungi. The author should clarigy this point, as the authors define the HGTs as interkindom HGTs.
(4) The authors did not cite the following reference. I suggest it should cited in this study. Liu F, Wang SH, Cheewangkoon R,and Zhao RL. 2025. Uneven distribution of prokaryote-derived horizontal gene transfer in fungi: a lifestyle-dependent phenomenon. mBio 16: https://doi.org/10.1128/mbio.02855-24
Comments on the Quality of English Language
No more comments
Author Response
The reviewer's detailed evaluation is acknowledged. Below is a point-by-point response to the concerns raised, with the corresponding changes highlighted in yellow in the revised manuscript.
(1) in the abstract, the authors should describe the study and results with past tense, instead of present tense.
We appreciate your feedback and assistance in refining the manuscript. The revisions have been incorporated into the current version of the paper.
(2) I wonder why the authors used proteomes data to search HGT, rather than the genome data.
Thank you for your insightful question. In our investigation, we opted to analyze proteome data rather than genomic data based on several key factors:
- Minimizing Interference from Non-Coding Regions: Genome sequences typically include large amounts of non-coding DNA, as well as repetitive and transposable elements, which can obscure the detection of horizontal gene transfer events. By focusing on proteome data, we effectively sidestep much of this extraneous information, concentrating on protein-coding sequences that are more likely to maintain the evolutionary signals necessary to identify HGT between unrelated species.
- Preservation of Evolutionary Signals: Protein sequences are generally more conserved than DNA sequences, which facilitates the detection of homologous relationships among distantly related species. This attribute is particularly valuable for pinpointing interkingdom HGT events, where significant sequence divergence often exists between donor and recipient organisms. Furthermore, the enhanced conservation of protein sequences bolsters the accuracy of phylogenetic reconstructions, a critical factor in differentiating genuine HGT events from other evolutionary processes.
- Alignment with Prior Research: Numerous studies investigating HGT in eukaryotes, particularly in fungi (e.g., Marcet-Houben & Gabaldón, 2010; Fitzpatrick, 2011), have employed proteome data for similar reasons. By adopting this established methodology, we ensure that our results are comparable to previous findings and can be seamlessly integrated into the broader context of HGT research.
(3) The study did not clarify the origin on the HGTs in Pezizomycotina fungi. The author should clarigy this point, as the authors define the HGTs as interkindom HGTs.
We appreciate the reviewer's helpful comment. To further clarify the origins of the putative HGT candidates in Pezizomycotina fungi, we have included this information in the supplementary material. Specifically, Tables S3 ("Patterns observed in phylogenetic analysis for Darkhorse HGT Candidates") and S4 ("Patterns observed in phylogenetic analysis of bibliographic candidates") summarize the origins of both the accepted and dismissed HGT candidates, based on bibliographic exploration and BLAST searches. Moreover, the origin of the eight identified HGT candidates is detailed in the manuscript—particularly in the Discussion section (lines 227-228) and further explained in the Results section (lines 159-160).
(4) The authors did not cite the following reference. I suggest it should cited in this study. Liu F, Wang SH, Cheewangkoon R,and Zhao RL. 2025. Uneven distribution of prokaryote-derived horizontal gene transfer in fungi: a lifestyle-dependent phenomenon. mBio 16: https://doi.org/10.1128/mbio.02855-24
We appreciate the reviewer's suggestion in bringing this interesting paper to our attention. We have now incorporated it into the revised manuscript (line 38, reference 10).
Reviewer 2 Report
Comments and Suggestions for Authors
Horizontal gene transfer(HGT)across species may be an important factor driving biological evolution and maintaining biodiversity.
More than 600 articles have been published indicating the existence of HGT events in the fungal subphylum Pezizomycotina.
The authors conducted a re-analysis of database data and literature, and found that HGT events in this particular species are not fully confirmed, and the probability of confirmed HGT events is extremely low.
The scientific problem that Ms attempts to explore is meaningful, but there are some major problems.
1. The title should reflect the exact results or ideas of the Ms;
2. The amount of data in the Ms is very limited, and the data analysis and literature sorting are also very preliminary;
3. The Ms needs to expand the amount of data and more analytical methods to prove such a large target.
Reviewer 3 Report
Comments and Suggestions for Authors
The study entitled “Rampant Interkingdom Horizontal Gene Transfer in Pezizomycotina? An Updated Inspection of Anomalous Phylogenies” is a very interesting one, with 49 bibliographic references, however there are some changes to be made:
Please exemplify more clearly the purpose of this manuscript, in which you list what you intend to research.
I don't recommend using "we" (e.g. line 292, line 294) - please rephrase. I find it helpful for readers to find an outline of all the methods used in this study, as described in the Supplementary files.
Figure 2 in the manuscript is not clear, please adapt it.
Good luck!
Author Response
The reviewer's detailed evaluation is acknowledged. Below is a point-by-point response to the concerns raised, with the corresponding changes highlighted in yellow in the revised manuscript.
(1) Please exemplify more clearly the purpose of this manuscript, in which you list what you intend to research.
Thank you for your insightful feedback. In response, we have revised the final paragraph of the introduction to more clearly articulate the purpose and scope of our research.
(2) I don't recommend using "we" (e.g. line 292, line 294) - please rephrase. I find it helpful for readers to find an outline of all the methods used in this study, as described in the Supplementary files.
We appreciate your feedback. In response, we have reworded the sentences containing "we" for improved clarity. Additionally, we have moved Figure S1 from the supplementary materials into the main document to enhance readers' understanding of the methods.
(3) Figure 2 in the manuscript is not clear, please adapt it.
Thank you for your observation. In this updated version of the manuscript, you'll find a figure featuring larger labels and enhanced resolution.
Round 2
Reviewer 2 Report
Comments and Suggestions for Authors
The authors re-present more data to support their views, I think the manuscript is acceptable. The author provided a new paper title, I think More suitable.